# MOBA: Model-Based Offline Reinforcement Learning with Adaptive Contextual Penalties

## Abstract

Mainstream model-based offline reinforcement learning, which aims to learn effective policies from static datasets, often employs conservatism to prevent policies from exploring out-of-support regions. For example, MOPO penalizes rewards through uncertainty measures from predicting the next states. Prior context-based approaches use meta-learning to infer latent dynamics, enabling a policy to adapt its behavior when deployed in out-of-support regions. This offers the potential for more robust decision-making compared to traditional model-based methods. However, current adaptive policy learning methods still leverage traditional conservative penalties to mitigate the compounding error of the model, which can overly constrain policy exploration. In this paper, we propose **MO**del-**B**ased Offline Reinforcement Learning with **A**daptive Contextual Penalty (MOBA), which introduces a context-aware penalty adaptation mechanism that dynamically adjusts conservatism based on trajectory history. Theoretically, we prove that MOBA maximizes a tighter lower bound on the true return compared to methods with fixed conservative penalties, achieving a more effective trade-off between risk and generalization. Empirically, we demonstrate that MOBA outperforms state-of-the-art model-based and model-free approaches on NeoRL and d4rl benchmark tasks. Our results highlight the importance of adaptive uncertainty estimation in model-based offline RL.

## 1 Introduction

Reinforcement learning (RL) has demonstrated remarkable success in domains where agents can learn through active trial-and-error interaction with an environment Sutton et al. (1998); Wang et al. (2018); Zhao et al. (2018); Shi et al. (2018). However, many real-world applications in areas like robotics, healthcare, and city management require learning from pre-existing, static datasets, as online exploration can be costly, unsafe, or impractical Luo et al. (2024); Zhang et al. (2019); Zhou et al. (2020); Vázquez-Canteli et al. (2019). Offline reinforcement learning Lange et al. (2012)Hein et al. (2017) Levine et al. (2020) Siegel et al. (2020) Jiang et al. (2015) Kumar et al. (2019)learns a policy directly from a pre-recorded dataset, thereby enabling a safer training paradigm compared to conventional online RL methods. Within offline RL, model-based approaches Wang et al. (2019); Kidambi et al. (2020); Yu et al. (2020)have gained significant attention. These methods explicitly learn a dynamics model from the offline dataset and subsequently use it to generate synthetic trajectories or to perform planning, thus improving data efficiency and enabling policy evaluation and optimization without further interaction with the environment. Model-based offline RL holds the promise of mitigating the sample inefficiency that plagues model-free methods, while also offering the potential for better generalization through model-based reasoning. Nevertheless, its performance is fundamentally limited by distributional shift, where errors in the learned model compound during policy optimization, leading to catastrophic overestimation of returns in out-of-support regions Fujimoto et al. (2019).

Previous model-based approaches address this challenge by penalizing rewards based on model uncertainty estimates. For example, MOPO Yu et al. (2020) penalizes rewards through the aleatoric uncertainty from predicting the next states, while MOBILE Sun et al. (2023) quantifies uncertainty via Bellman inconsistency. Although effective, these methods employ context-agnostic penalties that penalize all out-of-support transitions, limiting the potential of leveraging dynamics models. Access to states and actions outside the support region is more likely to be constrained by suppres-

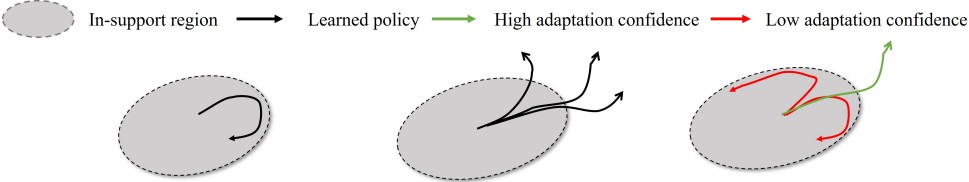

(a) model-based conservative methods    (b) context-aware meta-learning methods    (c) Our adaptive contextual penalty methods

Figure 1: A depiction of our method compared with previous methods. In Figure 1(b) and 1(c), multiple policy trajectories are shown because the meta-learning policy adapts its behavior based on the inferred dynamics model.

sion, thereby restricting the learned policy to confine the agent within regions similar to those of the behavioral policy.

In contrast, context-aware offline RL methods Rakelly et al. (2019) Chen et al. (2023) has emerged that leverage meta-learning to infer latent dynamics patterns and guide policy adaptation. By extracting latent representations of dynamics from prior experience, these methods enable policies to adapt more effectively in out-of-support regions during deployment. This approach allows for more robust decision-making compared to strictly conservative strategies, since it facilitates controlled exploration rather than overly restricting it. Nevertheless, despite employing context-aware policies to generalize beyond the training distribution, such methods typically still rely on reward penalties which leads to an inherently conservative bias that can limit overall performance.

In this work, we introduce MOBA, a novel model-based offline reinforcement learning framework that incorporates an adaptive contextual penalty mechanism to address the limitations of existing methods. Unlike traditional approaches that rely on context-agnostic uncertainty penalties to mitigate model errors in out-of-support regions, MOBA dynamically adjusts conservatism based on trajectory history. The proposed adaptive penalty mechanism operates through two synergistic components: a **context-recognition estimator** that quantifies the confidence of the policy given the inferred context and a **model-coverage estimator** that estimates the discrepancy between the real environment and the model environment. The difference between the aforementioned model-based methods and MOBA is shown in Figure1. In contrast to prior methods that either constrain the policy or use fixed penalties for adaptable policies, our method dynamically adjusts the penalty learning high adaptation confidence policy in out-of-support regions as well as constraining low adaptation confidence policies to out-of-support regions. Theoretically, we prove that MOBA's adaptive penalty tightens the lower bound on the true return compared to MOPO's context-agnostic baseline. This improvement stems from our method's ability to distinguish context-based uncertainty, a capability absent in prior work. Empirically, we evaluate MOBA on NeoRL and d4rl benchmark. Our resulting policy yields better performance than SOTA algorithms on 11 out of the 15 tasks. It is particularly noteworthy that MOBA surpassed MAPLE Chen et al. (2023) , which also employs an adaptable policy architecture, by and on two benchmarks seperatly. This demonstrates that under the adaptable policy architecture framework, the context-aware penalty coefficient method proposed in MOBA can effectively enhance algorithmic performance.

## 2 RELATED WORK

**Model-free offline RL**. Model-free offline RL methods constrain policies to the dataset's support to avoid exploiting out-of-distribution (OOD) states. Techniques include conservative Q-learning (CQL) Kumar et al. (2020), which penalizes Q-values for OOD actions, and TD3+BC Fujimoto & Gu (2021), which regularizes policies toward the dataset's behavior. While effective, these methods are inherently limited by the dataset's coverage for the learned policies are usually conservative since the dataset itself always limits the appropriate generalization of the learning policy beyond the offline dataset. Wang et al. (2021) Peng et al. (2019) Kostrikov et al. (2021) Xu et al. (2020)

**Model-based offline RL (MBRL)**. Model-based offline RL addresses these limitations by learning a dynamics model from the dataset to generate synthetic transitions Xu et al. (2018) Janner et al. (2019) Sun et al. (2018). Early approaches like MOPO Yu et al. (2020) penalize rewards using

model uncertainty estimates, while MOBILE Sun et al. (2023) quantifies uncertainty via Bellman inconsistency. The most recent works focus on designing better conservative strategies Kidambi et al. (2020) Rigter et al. (2022) Yu et al. (2021) Lu et al. (2021) to unleash the full potential of the model. MOREC Luo et al. (2023) learns a generalizable dynamics reward function from offline data, which is subsequently employed as a transition filter in any offline MBRL method. However, existing methods face compounding model errors, especially when policies explore OOD regions during long rollouts. MAPLE Chen et al. (2023) introduced contexture meta-policy learning in models to enable generalization to unseen situations.

## 3 PRELIMINARIES

**Markov Decision Processes and Offline RL** We consider a Markov Decision Process (MDP) defined by the tuple $\mathcal{M} = (\mathcal{S}, \mathcal{A}, T, r, \mu_0, \gamma)$ with state space $\mathcal{S}$, action space $\mathcal{A}$, transition dynamics $T(s'|s,a)$, reward function $r : \mathcal{S} \times \mathcal{A} \to \mathbb{R}$, initial state distribution $\mu_0$, and discount factor $\gamma \in (0,1)$. In offline RL, the agent learns a policy $\pi(a|s)$ from a static dataset $\mathcal{D} = \{(s_i, a_i, r_i, s_i')\}_{i=1}^N$ collected by an unknown behavior policy $\pi_\beta$, without environment interaction.

The objective is to maximize the expected return: Sutton et al. (1999)

$$\eta_M(\pi) = \mathbb{E}_{\substack{s_0 \sim \mu_0 \\ a_t \sim \pi(\cdot|s_t) \\ s_{t+1} \sim T(\cdot|s_t, a_t)}} \left[ \sum_{t=0}^{\infty} \gamma^t r(s_t, a_t) \right] \tag{1}$$

**Adaptable Policy Learning** Adaptable policy learning constructs an ensemble of plausible dynamics models $\{\hat{\rho}_i\}$ to represent all possible transition patterns in out-of-support regions. A meta-learned, context-aware policy leverages trajectory history to infer a latent environment context, dynamically adjusting its behavior to the inferred dynamics. The **environment context** is a latent vector, $z \in \mathcal{Z}$, that aims to capture the specific characteristics of an underlying dynamics model $\hat{\rho} \in \{\hat{\rho}_i\}$. This is achieved using an **environment-context extractor**, denoted by the mapping $\phi : \mathcal{T} \to \mathcal{Z}$, which infers the context $z$. In practice, this extractor is modeled as a recurrent neural network (RNN) that processes the history of interactions. At each timestep $t$, it updates the context based on the current state $s_t$, the previous action $a_{t-1}$, and the previous context $z_{t-1}$, according to the relation $z_t = \phi(s_t, a_{t-1}, z_{t-1})$. The policy, now **context-aware**, is conditioned on both the state and this inferred context: $\pi(a|s, z)$. The extractor $\phi$ and the policy $\pi_\phi$ are trained jointly to maximize the expected return across the entire ensemble of dynamics models, guided by the objective function:

$$\phi^*, \pi_{\phi^*}^* = \arg\max_{\phi, \pi_\phi} \mathbb{E}_{\hat{\rho} \sim \mathcal{T}}[J_{\hat{\rho}}(\pi_\phi)] \tag{2}$$

This optimization ensures that the extractor learns to produce informative contexts and that the policy learns to utilize these contexts to adapt its behavior effectively to any of the dynamics models it might encounter. During deployment, the agent engages in a probing and reducing phase: it executes actions in uncertain regions, collects real transition data, and iteratively refines its belief about the true dynamics until it converges to a single, well-adapted policy, ensuring safe generalization beyond the dataset's support.

## 4 METHODS

### 4.1 REFLECTION ON ADAPTABLE POLICY LEARNING'S LIMITATIONS

Adaptable policy learning, as exemplified by frameworks like MAPLE Chen et al. (2023), offers a principled approach to offline reinforcement learning by enabling policies to infer and adjust to uncertain dynamics in out-of-support regions during deployment. However, its practical efficacy is inherently limited by two key factors: the finite ensemble of learned dynamics models and the generalization capacity of the context extractor. To formalize these constraints, we introduce the probing-reducing paradigm, the framework that decomposes the deployment process into distinct phases and identifies the resultant error sources.

In this paradigm, policy deployment begins with a probing phase, where the policy explores a set of potential behaviors corresponding to the ensemble of dynamics models, gathering observations to infer the true underlying dynamics. This is followed by a reducing phase, where incompatible models are progressively eliminated, narrowing the policy set toward the optimal behavior under the inferred dynamics. The context extractor plays a central role in this reduction, iteratively encoding observed transitions into a latent context $z$, which clusters consistent dynamics models and guides policy adaptation.

Formally, consider an oracle context $z^*$ that modulates the policy $\pi(a \mid s, z^*)$ to match the optimal policy $\pi^*_{\widehat{T}}(a \mid s)$ under the closest model $\widehat{T}$ in the ensemble to the ground-truth dynamics $T^*$:

$$\widehat{T} = \arg\min_{T \in \mathcal{T}} \mathbb{E}_{(s,a) \sim \rho^\pi} \left[ D_{\mathrm{KL}} \left( T^*(\cdot \mid s, a) \| T(\cdot \mid s, a) \right) \right],$$

where $\mathcal{T}$ is the ensemble of models, and $\rho^\pi$ denotes the state-action occupancy under policy $\pi$. During deployment, a critical transition horizon $N_p$ exists such that for timesteps $k \geq N_p$, the inferred context $z_k$ converges sufficiently to $z^*$, reducing the policy set to the optimal one:

$$\pi(a \mid s, z_k) \approx \pi(a \mid s, z^*), \quad \forall k \geq N_p.$$

This paradigm reveals two primary error sources in out-of-support operations: (1) probing-phase recognition error, arising from delays in accurate context inference due to the extractor's limited generalization, which prolongs suboptimal actions during probing; and (2) dynamics-gap error, stemming from the ensemble's incomplete coverage of the continuous dynamics manifold, quantified by the minimal KL divergence between $T^*$ and the closest model in $\mathcal{T}$. Stronger context recognition accelerates reduction, minimizing probing error, while denser ensemble sampling reduces the dynamics gap.

This analysis highlights a need to rethink uncertainty penalization in offline model-based RL. Traditional methods apply static penalties based solely on model prediction errors, which rigidly constrain exploration and fail to account for adaptive inference. In contrast, our approach introduces a context-aware mechanism that dynamically modulates penalties based on probing efficiency and model discrepancy, enabling more robust adaptation.

### 4.2 CONTEXT-AWARE PENALTY IN OFFLINE MODEL-BASED ADAPTABLE POLICY LEARNING

Motivated by the probing-reducing paradigm, we derive a novel context-aware uncertainty penalization mechanism that characterizes the performance gap between policy execution in the true environment and the learned models. This represents a significant innovation in adaptable offline RL, as we are the first to integrate contextual adaptation directly into the penalty formulation, yielding a tighter theoretical bound on true returns compared to prior static approaches. We begin with a lemma establishing the relationship between policy returns under ground-truth dynamics $T^*$ and model dynamics $\widehat{T}$.

**Lemma 4.1** (Context-based Telescoping Lemma). *Let $M$ and $\widehat{M}$ be two MDPs sharing the reward function $r$ but differing in dynamics $T$ and $\widehat{T}$. Define the one-step model error as*

$$G^\pi_{\widehat{M}}(s, a) := \mathbb{E}_{s' \sim \widehat{T}(\cdot|s,a)} \left[ V^\pi_M(s') \right] - \mathbb{E}_{s' \sim T(\cdot|s,a)} \left[ V^\pi_M(s') \right].$$

*Then,*

$$\eta_{\widehat{M}}(\pi) - \eta_M(\pi) = \sum_{j=0}^{\infty} \gamma^{j+1} \mathbb{E}_{s_j, a_j \sim \pi, \widehat{T}} \left[ \lambda(s_j, a_j) G^\pi_{\widehat{M}}(s_j, a_j) \right],$$

*where*

$$0 \leq \lambda(s, a) = 1 - \epsilon(s, a)\omega(s, a) \leq 1.$$

Here, $\eta_M(\pi)$ denotes the expected return of policy $\pi$ in MDP $M$, $V^\pi_M(s)$ is the value function, and $\gamma$ is the discount factor. The coefficient $\lambda(s, a)$ decomposes the performance gap into two interpretable components, each addressing one of the error sources identified in the probing-reducing paradigm:

- $\epsilon(s, a)$ represents the context extractor's efficiency in recognizing the true dynamics during the probing phase. Specifically, it approximates the probability that the inferred context $z_k$ has converged to the oracle context $z^*$ (i.e., $k \geq N_p$, where $N_p$ is the transition horizon). A higher $\epsilon$ value indicates that the policy has quickly reduced its set of potential behaviors to the optimal one, thereby minimizing suboptimal actions taken while "probing" the environment. In essence, $\epsilon$ captures how well the context extractor generalizes from observed transitions to eliminate incompatible models early, directly mitigating probing-phase recognition error.

- $\omega(s, a) = 1 - \kappa(s, a)$, where $\kappa(s, a)$ quantifies the relative accuracy of the closest individual model in the ensemble (post-context inference) compared to the full ensemble average in approximating the ground-truth dynamics $T^*$. Derived from a ratio of expected value errors (see Appendix A), $\kappa(s, a)$ is smaller when the selected model closely matches $T^*$, leading to a lower $\omega$ and thus a smaller dynamics-gap error. This term reflects the irreducible discrepancy due to the ensemble's finite coverage of the dynamics space—lower $\omega$ means the inferred model is a strong approximation, reducing the risk of persistent errors even after probing.

**Theoretical advantage analysis** Under the assumption of Gaussian-distributed predictions from the ensemble models—a standard practice in model-based RL for quantifying uncertainty through predictive variance —the approximation error of the closest individual model is at most that of the full ensemble average. This implies $\kappa(s, a) \leq 1$, thereby ensuring the bound $0 \leq \lambda(s, a) \leq 1$ holds (see Appendix A for proof). To operationalize this, we adopt an admissible error estimator $u(s, a)$ that upper-bounds the model error: $u(s, a) \geq |G_{\widehat{M}}^{\pi}(s, a)|$. We use the max aleatoric error

$$u_{MOPO}(s, a) = \max_i \left\| \Sigma_\theta^i(s, a) \right\|_F$$

where $\left\{ \Sigma_\theta^i(s, a) \right\}_{i=1}^N$ are the variance heads of the ensemble models. We define the context-aware penalty as $p(s, a) = \lambda(s, a) \cdot u_{MOPO}(s, a)$, yielding the uncertainty-penalized reward $\tilde{r}(s, a) = r(s, a) - p(s, a)$. The corresponding penalized MDP is $\widetilde{M} = (\mathcal{S}, \mathcal{A}, \widehat{T}, \tilde{r}, \mu_0, \gamma)$. We use $\eta_M(\pi)$ to represent the expected discounted return (i.e., the long-term cumulative reward) of policy $\pi$ when executed in the true Markov Decision Process (MDP) $M$. This is the actual performance metric in the ground-truth environment, defined as: $\eta_M(\pi) = \mathbb{E}_{\pi, T} \left[ \sum_{t=0}^{\infty} \gamma^t r(s_t, a_t) \right]$, where $T$ is the true transition dynamics, $\gamma \in (0, 1)$ is the discount factor, and the expectation is over trajectories generated by starting from $\mu_0$, taking actions from $\pi$, and transitioning via $T$.

To compare our adaptive contextual penalty method with mainstream conservative methods, we use $\eta_{\text{MOPO}}(\pi)$ to represent the expected discounted return of policy $\pi$ in MOPO's penalized MDP, defined as: $\eta_{\text{MOPO}}(\pi) = \mathbb{E}_{\pi, \widehat{T}, \mu_0} \left[ \sum_{t=0}^{\infty} \gamma^t \tilde{r}_{\text{MOPO}}(s_t, a_t) \right]$ where $\tilde{r}_{\text{MOPO}}(s_t, a_t) = r(s_t, a_t) - u_{\text{MOPO}}(s_t, a_t)$ is the penalized reward, Similarly, $\eta_{\widetilde{M}}(\pi)$ represents the expected discounted return of policy $\pi$ when executed in the uncertainty-penalized MDP $\widetilde{M}$, defined as: $\eta_{\widetilde{M}}(\pi) = \mathbb{E}_{\pi, \widehat{T}} \left[ \sum_{t=0}^{\infty} \gamma^t \tilde{r}(s_t, a_t) \right]$

**Theorem 4.1** (Adaptive Penalty Lower Bound). *For any policy $\pi$, the true return is lower bounded by:*

$$\eta_M(\pi) \geq \eta_{\widetilde{M}}(\pi) \geq \eta_{MOPO}(\pi) \tag{3}$$

The proof, based on Lemma 4.1 and the triangle inequality, is provided in Appendix A. This bound is strictly tighter than those from static penalties, as $\lambda(s, a) \leq 1$ adapts to contextual confidence, reducing conservatism in regions where probing has succeeded or the dynamics gap is small.

## 4.3 PRACTICAL IMPLEMENTATION

We implement this framework by extending ensemble-based dynamics modeling with the adaptive penalty. An ensemble of $m$ probabilistic models $\{\widehat{T}_i\}_{i=1}^m$ is trained via supervised learning on the offline dataset $\mathcal{D}$, where each model predicts $\widehat{T}_i(s' \mid s, a) \sim \mathcal{N}(\mu_i(s, a), \Sigma_i(s, a))$ under the Gaussian assumption. A recurrent context extractor $\psi_\xi(z_t \mid s_t, a_{t-1}, z_{t-1})$ infers the context variable $z_t$ from trajectory history.

The policy $\pi_\theta(a \mid s_t, z_t)$ conditions on both the state $s_t$ and context $z_t$. The penalty term is defined as $p_t = \lambda(s_t, a_t) \cdot u(s_t, a_t)$, with $u(s_t, a_t)$ as defined above.

For $\epsilon(s_t, a_t)$, we use the normalized entropy of the action distribution:

$$\epsilon(s_t, a_t) = 1 - H(\pi_\theta(\cdot \mid s_t, z_t)),$$

where higher confidence (lower entropy) indicates the post-probing phase.

For $\omega(s_t, a_t)$, we define:

$$\omega(s_t, a_t) = \frac{\text{Var}_i[\mu_i(s_t, a_t)]}{\mathbb{E}[|\Sigma_i(s_t, a_t)|]},$$

This ratio captures optimal model discrepancy by emphasizing inter-model disagreement (numerator) relative to average intra-model uncertainty (denominator). It better quantifies the dynamics gap than raw variance, as it normalizes for intrinsic model noise, highlighting regions where the closest model deviates significantly from the ensemble mean—indicative of poor approximation to ground truth. Training proceeds as in Algorithm 1, using SAC to update $\theta$ and $\xi$ on $\mathcal{D} \cup \mathcal{D}_{\text{rollout}}$, with penalized rewards in rollouts.

---

**Algorithm 1** MOBA: Model-Based Offline RL with Adaptive Contextual Penalty

---

1: **Input**: Offline dataset $\mathcal{D}$, ensemble size $m$, rollout horizon $H$
2: Initialize ensemble $\{\widehat{T}_i\}_{i=1}^m$, policy $\pi_\theta$, context extractor $\psi_\xi$, rollout buffer $\mathcal{D}_{\text{rollout}} \leftarrow \emptyset$
3: **for** each training iteration **do**
4:     **Rollout Generation:**
5:     **for** $k = 1$ to $K$ **do**
6:         Sample initial state $s_0 \sim \mathcal{D}$, select $\widehat{T}_i \sim \{\widehat{T}_i\}_{i=1}^m$
7:         **for** $t = 0$ to $H - 1$ **do**
8:             Compute context $z_t \leftarrow \psi_\xi(z_t \mid s_t, a_{t-1}, z_{t-1})$
9:             Sample action $a_t \sim \pi_\theta(a \mid s_t, z_t)$
10:           Predict next state $s_{t+1} \sim \widehat{T}_i(s' \mid s_t, a_t)$
11:           Compute reward $r_t \leftarrow r(s_t, a_t)$
12:           Compute $\epsilon(s_t, a_t), \omega(s_t, a_t)$
13:           Compute $\lambda(s_t, a_t) \leftarrow 1 - \epsilon(s_t, a_t) \cdot \omega(s_t, a_t)$
14:           Adjust reward $r'_t \leftarrow r_t - \lambda(s_t, a_t) \cdot u(s_t, a_t)$
15:           Store $(s_{t+1}, r'_t, d_{t+1}, s_t, a_t, z_t)$ in $\mathcal{D}_{\text{rollout}}$
16:           **if** termination **then**
17:              **break**
18:           **end if**
19:         **end for**
20:     **end for**
21:     **Parameter Updates:**
22:     Update $\theta, \xi$ using SAC on $\mathcal{D} \cup \mathcal{D}_{\text{rollout}}$ to maximize expected return
23: **end for**
24: **Output**: Optimized policy $\pi_\theta$, context extractor $\psi_\xi$

---

## 5 EXPERIMENTS

We assess the effectiveness of the adaptive contextual penalty mechanism in enhancing policy performance, particularly in scenarios requiring generalization to out-of-support regions. In this section, we first compare MOBA against state-of-the-art model-based and model-free offline RL methods to answer how does MOBA compare to previous methods in standard offline RL benchmarks. We then conduct ablation studies to isolate the impact of the adaptive penalty, thereby demonstrating the critical role of our proposed dynamic penalty coefficient in effectively balancing constraint satisfaction and optimization performance throughout the training process. Next, we will present a demonstration by tracking the performance of parameters during the training process to justify our selection of $\epsilon(s, a), \omega(s, a)$. Also, we present a comparative analysis of uncertainty penalties across different contexts to demonstrate how our context-based penalty method, through its integration with environmental identification, yields penalty estimation that are better aligned with adaptable policy

requirements. Finally, we will demonstrate the performance variation of the MOBA algorithm with the increase of the rollout horizon to a certain extent. Traditional adaptable policies exhibit performance degradation when the rollout horizon is extended, which limits the algorithm's capability. We will show that our algorithm mitigates this issue to some degree and provide analyses to the mechanism behind this improvement. During the submission period, we temporarily hosted the code on an anonymous GitHub repository.[1].

## 5.1 BENCHMARK RESULTS

**Comparative Evaluation on NeoRL Benchmark**. We evaluate MOBA on on the NeoRL benchmark Qin et al. (2021) Gao et al. (2025). We compare MOBA against six baseline robot locomotion control tasks. Table 1 shows normalized scores. MOBA outperforms previous SOTA methods in most of the tasks and achieves the highest average score among all methods. The success achieved in the challenging NeoRL benchmark strongly demonstrates the potential of our algorithm in real-world scenarios. The detailed baselines and hyper-parameters are listed in AppendixB

Table 1: Performance comparison on NeoRL tasks. Normalized scores (mean $\pm$ standard deviation) are reported at the final training iteration. Bold denotes the best mean performance.

| Task Name | BC | CQL | TD3+BC | EDAC | MOPO | MOBILE | MAPLE | MOBA |
|---|---|---|---|---|---|---|---|---|
| HalfCheetah-L | 29.1 | 38.2 | 30.0 | 31.3 | 40.1 | **54.7** | 36.2 | $51.3 \pm 0.37$ |
| Hopper-L | 15.1 | 16.0 | 15.8 | 18.3 | 6.2 | 17.4 | 22.7 | $\mathbf{32.97} \pm \mathbf{0.27}$ |
| Walker2d-L | 28.5 | 44.7 | 43.0 | 40.2 | 11.6 | 37.6 | 33.8 | $\mathbf{70.46} \pm \mathbf{0.77}$ |
| HalfCheetah-M | 49.0 | 54.6 | 52.3 | 54.9 | 62.3 | 77.8 | 75.5 | $\mathbf{86.2} \pm 1.32$ |
| Hopper-M | 51.3 | 64.5 | 70.3 | 44.9 | 1.0 | 51.1 | 27.7 | $\mathbf{74.56} \pm 18.3$ |
| Walker2d-M | 48.7 | 57.3 | 58.5 | 57.6 | 39.9 | 62.2 | 40.7 | $\mathbf{78.63} \pm 2.23$ |
| Average | 37.0 | 45.9 | 45.0 | 41.2 | 26.9 | 50.1 | 39.4 | **65.7** |

**Comparative Evaluation on d4rl Benchmark**. We evaluate MOBA on on the d4rl Fu et al. (2020) benchmark,which includes Gym and Adroit domains.Table 2 shows normalized scores. MOBA demonstrates superior or competitive performance across the majority of tasks. The results demonstrate that the policy induced by adaptive penalty coefficients outperforms that derived solely from the uncertainty measure associated with predicting the subsequent state.

Table 2: Performance comparison on d4rl tasks,Normalized scores (mean $\pm$ standard deviation) are reported at the final training iteration. Bold denotes the best mean performance.

| Task Name | CQL | TD3+BC | EDAC | MOPO | COMBO | TT | RAMBO | MOBILE | MAPLE | MOBA |
|---|---|---|---|---|---|---|---|---|---|---|
| hfctah-rnd | 31.3 | 11.0 | 28.4 | 38.5 | 38.8 | 6.1 | 39.5 | 39.3 | 38.4 | $38.3 \pm 1.23$ |
| hopper-rnd | 5.3 | 8.5 | 25.3 | 31.7 | 17.9 | 6.9 | 25.4 | 31.9 | 10.6 | $\mathbf{33.2} \pm 0.1$ |
| walker-rnd | 5.4 | 1.6 | 16.6 | 7.4 | 7.0 | 5.9 | 0.0 | 17.9 | 21.7 | $\mathbf{24.1} \pm 0.67$ |
| hfctah-med | 46.9 | 48.3 | 65.9 | 73.0 | 54.2 | 46.9 | 77.9 | 74.6 | 50.4 | $\mathbf{79.8} \pm 1.23$ |
| hopper-med | 61.9 | 59.3 | 101.6 | 62.8 | 97.2 | 67.4 | 87.0 | 106.6 | 21.1 | $105.6 \pm 0.1$ |
| walker-med | 79.5 | 83.7 | 92.5 | 84.1 | 81.9 | 81.3 | 84.9 | 87.7 | 56.3 | $\mathbf{92.2} \pm 2.06$ |
| hfctah-med-rep | 45.3 | 44.6 | 61.3 | 72.1 | 55.1 | 44.1 | 68.7 | 71.7 | 59.0 | $69.7 \pm 0.1$ |
| hopper-med-rep | 86.3 | 60.9 | 101.0 | 103.5 | 89.5 | 99.4 | 99.5 | 103.9 | 87.5 | $\mathbf{110.8} \pm 0.4$ |
| walker-med-rep | 76.8 | 81.8 | 87.1 | 85.6 | 56.0 | 82.6 | 89.2 | 89.9 | 76.7 | $\mathbf{95.1} \pm 3.7$ |
| Average | 48.7 | 44.4 | 64.4 | 62.1 | 55.3 | 49.0 | 63.6 | 69.3 | 46.9 | **72.5** |

## 5.2 ABLATION STUDIES

To evaluate the effectiveness of the adaptive penalty mechanism in MOBA, we conduct three targeted experiments. These studies isolate the contributions of the adaptive components and validate the design choices for $\epsilon(s, a)$ and $\omega(s, a)$, as well as the overall uncertainty estimation strategy.

---

[1]https://anonymous.4open.science/r/MOBA-D6AB

To quantify the contribution of adaptive penalties, we compare MOBA with variants with fixed penalty coefficients: 1) $\epsilon$ fixed to 1.0, 2) $\omega$ fixed to 1.0, and 3) both fixed to 1.0 scenarios. We select Walker2d-low to verify the argument. Figure 2 demonstrates that MOBA consistently outperforms these variants across training epochs. The adaptive adjustment of both $\epsilon$ and $\omega$ enables better handling of trajectory-dependent uncertainties, confirming the advantage of context-aware penalty adaptation over context-unaware approaches.

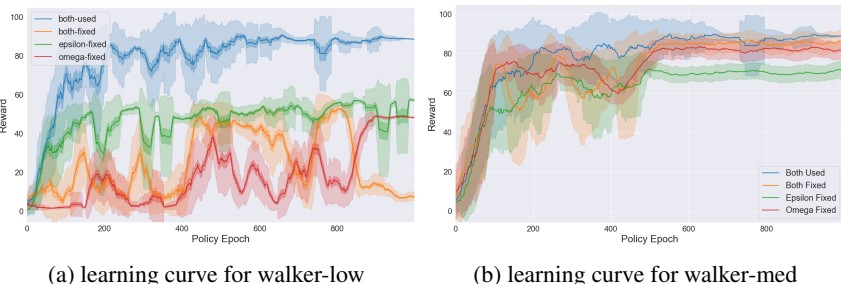

| (a) learning curve for walker-low | (b) learning curve for walker-med |

Figure 2: Uncertainty penalty compared over different context

Second, we justify the chosen formulations for $\epsilon(s,a)$ and $\omega(s,a)$ by analyzing their behaviors. For a properly selected $\epsilon(s,a)$, it is hypothesized that the value of $\epsilon(s,a)$ will monotonically increase during the rollout process. This theoretical prediction stems from the inherent property of $\epsilon(s,a)$ as an estimator of discriminative capability: as the rollout length extends, the divergence between distinct environmental trajectories amplifies, thereby enabling the policy to progressively enhance its discriminative capacity in environmental identification. Figure3a illustrates $\epsilon(s,a)$ across rollout steps averaged over 100 epochs. This metric reflects the policy's contextual awareness, showing a stable trend with minor fluctuations, validating its suitability as a probing phase recogniton error indicator.

For an appropriately chosen $\omega(s,a)$, we hypothesize that as the number of models increases, the average value of $\omega(s,a)$ during training remains relatively high. This is attributed to the fact that $\omega(s,a)$ serves as an estimator of model coverage capability, where a greater number of models inherently enhances the collective coverage capacity of the ensemble. Figure3b depicts $\omega(s,a)$, defined as the ratio of ensemble variance to expected covariance magnitude, across different model sizes. The analysis reveals that $\omega(s,a)$ decreases with larger ensembles, indicating improved dynamics approximation, thus supporting our choice of this metric for capturing dynamics gap error.

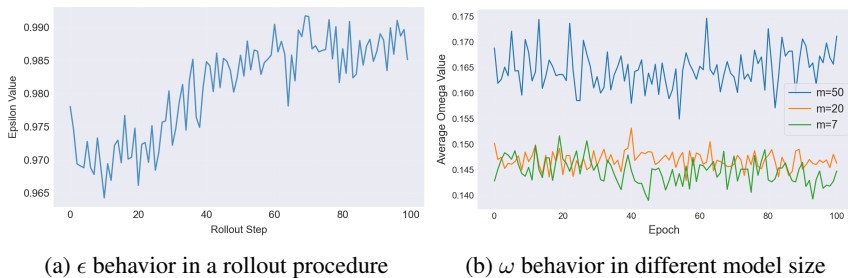

| (a) $\epsilon$ behavior in a rollout procedure | (b) $\omega$ behavior in different model size |

Figure 3: A depiction of the dynamic changes in parameters $\epsilon$ and $\omega$ throughout the training process

To verify the effectiveness of MOBA in terms of context recognition, we assess uncertainty estimation by comparing its behavior under 1) learned context 2) learned context with a gaussian noise perturbation 3) random context. All experiments are done in the walker2d-low environment, with same penalty coefficient. As shown in figure4. The uncertainty penalty remains lowest and most stable with real context, increases with noise perturbation, and exhibits the highest variability with random context. This confirms that the context-aware uncertainty estimator effectively leverages trajectory history. With a well learned context, the uncertainty penalty drops to avoid overly restriction on policy exploration.

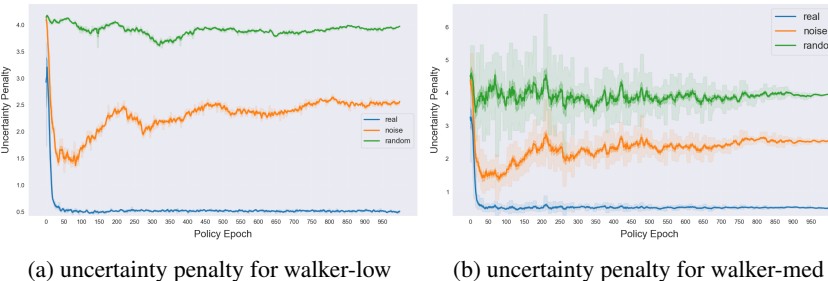

(a) uncertainty penalty for walker-low  (b) uncertainty penalty for walker-med

Figure 4: Uncertainty penalty compared over different context

In traditional adaptable policy learning methods, as reported by MAPLE, the asymptotic performance gradually deteriorates as the rollout horizon $H$ increases. This phenomenon stems from the progressive accumulation of compounding errors with increasing $H$, where the adaptable policy overfits the finite dynamics models and consequently fails to infer a correct environmental context for policy adaptation when deployed. In contrast, our methodology demonstrates a certain degree of performance improvement with a limited range of increment in $H$, as shown in Figure5 . This indicates that our algorithm can mitigate the overfitting issue of the adaptable policy to a certain extent. For deployment environments that significantly differ from the model environment, the policy exhibits poor contextual identification capability, while the model's coverage of the environment remains low. Such scenarios are constrained during training by a large uncertainty penalty. Consequently, our algorithm can tolerate a longer rollout horizon, thereby enhancing its overall performance.

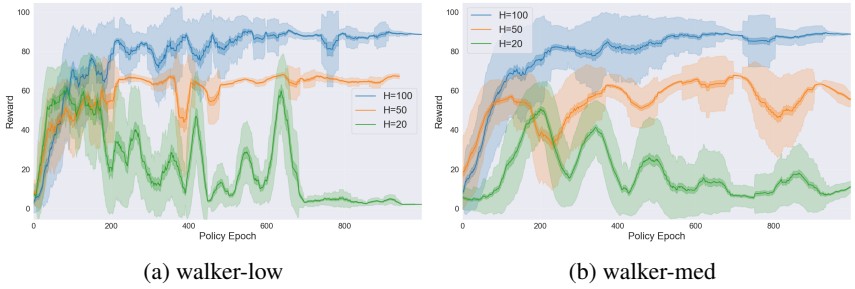

(a) walker-low  (b) walker-med

Figure 5: Performance Under different horizon

# 6 DISCUSSION AND FUTURE WORK

In this paper, we propose MOBA, a novel model-based offline reinforcement learning framework with an adaptive contextual penalty mechanism, which addresses the limitations of existing methods by dynamically adjusting conservatism based on trajectory history. Our theoretical analysis proves that MOBA maximizes a tighter lower bound on the true return compared to prior methods like MOPO, and empirical results on the NeoRL and d4rl benchmarks demonstrate its superior performance over state-of-the-art model-based and model-free approaches. These findings highlight the critical role of adaptive uncertainty estimation in enhancing policy generalization to out-of-support regions.

Despite these advancements, MOBA's practical implementation reveals significant scope for improvement. The choice for $\epsilon(s, a)$ and $\omega(s, a)$—currently based on empirical justification—could be refined through automated hyperparameter tuning or theoretical optimization. Although our ablation studies justify these choices, a principled derivation of $\epsilon(s, a)$ and $\omega(s, a)$ from first principles could further enhance robustness, potentially leading to a more generalized framework.

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

# A PROOFS

## A.1 PROBLEM FORMALIZATION

We formalize the problem under the Markov Decision Process framework, where the true environment is denoted as $M^* = (\mathcal{S}, \mathcal{A}, T^*, r, \mu_0, \gamma)$ with $T^*(s'|s,a)$ representing the ground-truth transition dynamics. To approximate this system, we employ an ensemble of probabilistic models $\{\widehat{T}_{\phi_i}(s'|s,a)\}_{i=1}^N$ that collectively form a mixture distribution $\widehat{T}_\phi(s'|s,a) = \frac{1}{N}\sum_{i=1}^N \mathcal{N}(\mu_{\phi_i}(s,a), \Sigma_{\phi_i}(s,a))$.

We denote $\widehat{T}_\theta(s'|s,a)$ as the model in the ensemble that is closest to the ground truth environment. A trainable context extractor $\psi_\xi(z|\tau_{0:t})$ generates latent context variables $z \in \mathcal{Z}$ from trajectory history $\tau_{0:t} = (s_0, a_0, ..., s_t)$, enabling temporal adaptation.

## A.2 PROOF OF TELESCOPING LEMMA

We now provide a proof for Lemma 4.1 from the main text.

**Lemma A.1** (Telescoping Lemma for Adaptive Model Switching). *The difference in returns between the learned policy in the model and the true environment can be expressed as:*

$$\eta_{\widehat{M}}(\pi) - \eta_{M^*}(\pi) = \sum_{j=0}^\infty \gamma^{j+1} \mathbb{E}_{s_j, a_j \sim \pi, \widehat{T}} \left[ \lambda(s_j, a_j) G_{\widehat{M}}^\pi(s_j, a_j) \right] \tag{4}$$

*where $\lambda(s,a)$ is an adaptive weighting function.*

*Proof.* We begin by defining the value difference function:

$$G_{\widehat{M}}^\pi(s,a) = \gamma \left( \mathbb{E}_{s' \sim \widehat{T}(s,a)} \left[ V_M^\pi(s') \right] - \mathbb{E}_{s' \sim T^*(s,a)} \left[ V_M^\pi(s') \right] \right) \tag{5}$$

Using the telescoping lemma from MOPO Yu et al. (2020), the performance difference can be expressed as a sum over horizon steps. In our adaptive framework, we employ different transition models at different stages:

$$\begin{aligned}
\eta_{\widehat{M}}(\pi) - \eta_{M^*}(\pi) = \sum_{j=0}^{N_p} \gamma^{j+1} \mathbb{E}_{s_j, a_j \sim \pi, \widehat{T}_\phi} \left[ G_{\widehat{M}}^\pi(s_j, a_j) \right] \\
+ \sum_{j=N_p+1}^\infty \gamma^{j+1} \mathbb{E}_{s_j, a_j \sim \pi, \widehat{T}_\theta} \left[ G_{\widehat{M}}^\pi(s_j, a_j) \right]
\end{aligned} \tag{6}$$

where $N_p$ represents the transition step where we switch from the ensemble model $\widehat{T}_\phi$ to the single best model $\widehat{T}_\theta$.

To unify these expressions, we apply importance sampling to relate the expectations under different models. First, we rewrite $G_{\widehat{M}}^\pi(s,a)$ using importance sampling:

$$G_{\widehat{M}}^\pi(s,a) = \gamma^{j+1} \mathbb{E}_{s_j, a_j \sim \pi, \widehat{T}} \left[ \mathbb{E}_{s' \sim T^*(s_j, a_j)} \left[ \left( \frac{\widehat{T}(s'|s_j, a_j)}{T^*(s'|s_j, a_j)} - 1 \right) V_M^\pi(s') \right] \right] \tag{7}$$

For the two transition models $\widehat{T}_\phi$ (ensemble) and $\widehat{T}_\theta$ (single best), we define their respective contribution functions:

$$\begin{aligned}
G_\phi(s_j, a_j) = \mathbb{E}_{s' \sim T^*(s_j, a_j)} \left[ \left( \frac{\widehat{T}_\phi(s'|s_j, a_j)}{T^*(s'|s_j, a_j)} - 1 \right) V_M^\pi(s') \right] \\
G_\theta(s_j, a_j) = \mathbb{E}_{s' \sim T^*(s_j, a_j)} \left[ \left( \frac{\widehat{T}_\theta(s'|s_j, a_j)}{T^*(s'|s_j, a_j)} - 1 \right) V_M^\pi(s') \right]
\end{aligned} \tag{8}$$

The relationship between expectations under different models can be expressed through the ratio:

$$\mathbb{E}_{s_j,a_j\sim\pi,\widehat{T}_\theta}\left[G^\pi_{\widehat{M}}(s_j,a_j)\right] = \mathbb{E}_{s_j,a_j\sim\pi,\widehat{T}_\phi}\left[\kappa(s_j,a_j)G^\pi_{\widehat{M}}(s_j,a_j)\right] \tag{9}$$

where

$$\kappa(s,a) = \frac{G_\theta(s,a)}{G_\phi(s,a)} = \frac{\mathbb{E}_{s'\sim T^*(s,a)}\left[\left(\frac{\widehat{T}_\theta(s'|s,a)}{T^*(s'|s,a)}-1\right)V^\pi_M(s')\right]}{\mathbb{E}_{s'\sim T^*(s,a)}\left[\left(\frac{\widehat{T}_\phi(s'|s,a)}{T^*(s'|s,a)}-1\right)V^\pi_M(s')\right]} \tag{10}$$

Combining both parts of the performance difference:

$$\eta_{\widehat{M}}(\pi) - \eta_{M^*}(\pi) = \sum_{j=0}^{N_p}\gamma^{j+1}\mathbb{E}_{s_j,a_j\sim\pi,\widehat{T}_\phi}\left[G^\pi_{\widehat{M}}(s_j,a_j)\right]$$

$$+ \sum_{j=N_p+1}^{\infty}\gamma^{j+1}\mathbb{E}_{s_j,a_j\sim\pi,\widehat{T}_\phi}\left[\kappa(s_j,a_j)G^\pi_{\widehat{M}}(s_j,a_j)\right] \tag{11}$$

$$= \sum_{j=0}^{\infty}\gamma^{j+1}\mathbb{E}_{s_j,a_j\sim\pi,\widehat{T}_\phi}\left[\lambda(s_j,a_j)G^\pi_{\widehat{M}}(s_j,a_j)\right]$$

where $\lambda(s,a)$ is the piecewise weighting function:

$$\lambda(s,a) = \begin{cases} 1, & j \le N_p \\ \kappa(s,a), & j > N_p \end{cases}$$

$\square$

### A.3 PROPERTIES OF THE ENSEMBLE MODEL

For a Gaussian ensemble model, the ratio term can be analyzed as follows:

$$\frac{\widehat{T}_\phi(s'|s,a)}{T^*(s'|s,a)} - 1 = \frac{\frac{1}{N}\sum_{i=1}^N\widehat{T}_{\phi_i}(s'|s,a)}{T^*(s'|s,a)} - 1 = \frac{1}{N}\sum_{i=1}^N\left(\frac{\widehat{T}_{\phi_i}(s'|s,a)}{T^*(s'|s,a)}\right) - 1 \tag{12}$$

Defining the error of a single model as $e_i(s') = \mathbb{E}_{s'\sim T^*(s,a)}\left[\frac{\widehat{T}_{\phi_i}(s'|s,a)}{T^*(s'|s,a)}-1\right]$, the ensemble error becomes $e_\phi(s') = \frac{1}{N}\sum_{i=1}^N e_i(s')$. Since $\widehat{T}_\theta$ is the model closest to the real dynamics, we have $e_\theta \le e_\phi$, which implies:

$$\kappa(s,a) \le 1 \tag{13}$$

### A.4 PRACTICAL APPROXIMATION

In practice, the exact transition step $N_p$ is unknown. We therefore introduce $\epsilon(s,a)$ as the estimated probability that $j \ge N_p$ (i.e., the probability that we have switched to the single model). The weighting function $\lambda(s,a)$ can then be approximated as:

$$\begin{aligned} \lambda(s,a) &= (1-\epsilon(s,a))\cdot 1 + \epsilon(s,a)\cdot\kappa(s,a) \\ &= 1 - \epsilon(s,a)(1-\kappa(s,a)) \\ &= 1 - \epsilon(s,a)\omega(s,a) \end{aligned} \tag{14}$$

where $\omega(s,a) = 1 - \kappa(s,a) \ge 0$ represents the relative advantage of the single model over the ensemble.

This formulation provides a practical way to implement the adaptive weighting scheme described in the main text.

### A.5   Proof of Adaptive Penalty Lower Bound

We now provide a proof of theorem 4.1

*Proof.* Start from telescoping decomposition:

$$\eta_M(\pi) = \eta_{\widetilde{M}}(\pi) - \mathbb{E}[\lambda(s,a)G^\pi_{\widetilde{M}}(s,a)] \tag{15}$$

Apply triangle inequality :

$$\begin{aligned}
\eta_M(\pi) &\geq \eta_{\widetilde{M}}(\pi) - \lambda(s,a)\mathbb{E}[u_{\text{MOPO}}(s,a)] \\
&= \mathbb{E}[r - \lambda(s,a)u_{\text{MOPO}}(s,a)] \\
&\geq \mathbb{E}[r - u_{\text{MOPO}}(s,a)] = \eta_{\text{MOPO}}(\pi)
\end{aligned} \tag{16}$$

$\square$

Theorem 4.1 show that our adaptive coefficient $\lambda_{\text{MOBA}} = 1 - \epsilon(s,a)\omega(s,a)$ provides a strictly tighter lower bound on the true return.

## B   Implementation Details

### B.1   Benchmark

Here we introduce the baselines used in our experiments, including model-free offline RL and model-based offline RL. The offline RL benchmarks include :

1. CQL Kumar et al. (2020), which penalizes Q-values on OOD samples

2. TD3+BC Fujimoto & Gu (2021), which adopts a BC constraint when optimizing policy

3. EDAC An et al. (2021),which leverages clipped Q-learning to penalize based on the uncertainty degree of the Q-value

4. COMBO Yu et al. (2021),which applies CQL in dyna-style enforces Q-values small on OOD samples

5. RAMBO Rigter et al. (2022),which trains the policy and the dynamics model adversarially

6. MOPO Yu et al. (2020),which learns a pessimistic value function from penalty-adjusted rewards due to uncertainty in dynamic model predictions.

7. MOBILE Sun et al. (2023),which penalizes the rewards with uncertainty quantified by the inconsistency of Bellman estimations

### B.2   hyperparameter

The hyper-parameters for MOBA is derived from the default parameters of MOBILE. The major modification we did is lengthen the rollout horizen to 100, for all experiments, and use a larger model ensemble. The details are listed in table 3

Table 3: Common Hyper-parameters of MOBA

| Attribute | Value |
|---|---|
| Actor learning rate | 3e-4 |
| Critic learning rate | 3e-4 |
| Dynamics learning rate | 1e-3 |
| Model ensemble size | 50 |
| Number of the selected models | 35 |
| The number of critics | 2 |
| Hidden layers of the actor network | [256, 256] |
| Hidden layers of the critic network | [256, 256] |
| Hidden layers of the dynamics model | [200, 200, 200, 200] |
| Discount factor $\gamma$ | 0.99 |
| Target network smoothing coefficient $\tau$ | $5 \times 10^{-3}$ |
| Max Rollout horizon $H_{\text{rollout}}$ | 100 |
| Optimizer of the actor and critics | Adam |
| Rollout number per epoch | 2000 |
| Batch size of optimization | 256 |
| Batch number of inferring reward | 4 |
| Total gradient steps | $1 \times 10^6$ |
| Actor optimizer learning schedule | Cosine learning schedule |

