# OpenReview forum: "MOBA: Model-Based Offline Reinforcement Learning with Adaptive Contextual Penalties"
_ICLR.cc/2026/Conference — Submitted to ICLR 2026_

### Official Review · Reviewer_sBH5 · 2025-10-28

**Soundness:** 3
**Presentation:** 2
**Contribution:** 2
**Rating:** 2
**Confidence:** 4

**Summary:**

This paper addresses the over-constraint issue of fixed conservative penalties in mainstream model-based offline RL. It proposes MOBA, a framework with a context-aware adaptive penalty mechanism that adjusts conservatism via trajectory history (using context-recognition estimator $\epsilon$ and model-coverage estimator $\omega$). Theoretically, MOBA tightens the lower bound on true returns compared to fixed-penalty methods like MOPO; empirically, it outperforms SOTA approaches on NeoRL and d4rl tasks.

**Strengths:**

1. The paper clearly analyzes probing-phase recognition error and dynamics-gap error in adaptive policy learning by splitting deployment into probing and reducing phases.
2. The paper’s theoretical analysis thoroughly proves the Context-based Telescoping Lemma and shows MOBA’s adaptive penalty tightens the true return lower bound, outperforming fixed-penalty methods like MOPO in theory.
3. The paper’s experimental analysis is comprehensive, using ablations to validate $\epsilon$ and $\omega$, and testing rollout horizon robustness, fully verifying the context-aware penalty’s effectiveness.

**Weaknesses:**

1. The design rationale for $\epsilon$ (representing probing-phase recognition error) is insufficiently clarified. The paper does not explicitly explain why using the normalized entropy of the action distribution can effectively reflect the context extractor’s error in identifying true dynamics during exploration.
2. Several equations in proofs in Appendix A contain ambiguities or inconsistencies. For example, the rewarite of $G_{\hat{M}}^\pi(s,a)$ in equation (7) is inconsistent with equation (5).
3. While the context-aware penalty quantification mechanism is valuable, the core innovation remains limited. It operates within the existing adaptive policy learning framework of model-based offline RL, with penalty designs rooted in MOPO, leading to modest overall novelty.

**Questions:**

1. Why is the normalized entropy of actions used to reflect the probing-phase error of the context extractor? From my understanding, The normalized entropy of actions only measures the policy’s certainty about the next action, but it cannot quantify the exploration error of the context accumulated from states prior to the current one.
2. Most of the benchmark methods compared in the paper were published before 2023. Is this because no new model-based reinforcement learning methods have been published in the last two to three years?

---

> ### Author Response · Authors · 2025-11-17
>
> We thank Reviewer sBH5 for the meticulous review and for acknowledging the strengths of our theoretical analysis, including the Telescoping Lemma and the tighter lower bound. We address your specific concerns below.
>
> W1, Q1:Why is normalized entropy used to reflect probing-phase error?
>
>  You are correct that policy entropy directly measures "the policy's certainty" and not directly "the context extractor's error". Our central argument is that, in our framework, the former is a principled and direct proxy for the latter. Our "probing-reducing paradigm" (Section 4.1) involves two coupled components: the context extractor $\psi_\xi$ and the context-aware policy $\pi_\theta(a|s,z)$.The goal of the extractor is to produce an informative context $z_t$.If the extractor fails (i.e., high recognition error, still in the "probing" phase), the context extractor fails to eliminate the incompatible models, the agent effectively faces a superposition of multiple possible dynamics. Since the optimal action often varies significantly across these conflicting dynamics, a rational policy cannot commit to a single action. Instead, it must maintain a dispersed probability mass (high entropy) to hedge against this uncertainty. Conversely, when the extractor succeeds (low recognition error, in the "reducing" phase), it provides a sharp, informative $z_t$. The policy can then confidently select the optimal action, exhibiting low policy entropy (a high $\epsilon$ value).Therefore, high entropy is a direct signature of the agent facing multiple possible environments due to an uninformative context. We will add this explicit line of reasoning to a new subsection in the revision.
>
> W2: Several equations in proofs in Appendix A contain ambiguities or inconsistencies .
>
> Thank you for this careful catch. You are right to point this out.Eq (5) vs (7): Equation (5) defines $G_{\hat{M}}^{\pi}(s, a)$. Equation (7) is a rewrite of this same term using importance sampling, which is a key step in the proof. We failed to add the necessary explanatory text. We will add a sentence explicitly stating: "We now rewrite $G_{\hat{M}}^{\pi}(s, a)$ using importance sampling with respect to the true dynamics $T^*$
>
> W3: Limited core innovation
>
> We would like to provide a more precise clarification that MOBA's contribution is a critical and novel synthesis that addresses a fundamental shortcoming in prior work. Prior attempts to leverage adaptive-policy-based approaches for offline RL have been hindered by the limited size of the ensemble, forcing the re-introduction of state-dependent uncertainty penalties—an expedient that directly contradicts the defining spirit of adaptive policies. We propose a practical remedy by augmenting the training objective with an adaptive penalty coefficient. This yields a computationally lightweight yet theoretically grounded solution that retains the full benefits of adaptive policy learning even when the ensemble is small. We believe that although our implementation is concise, its conceptual contribution is not means limited. While MOBA builds upon the adaptable policy learning framework introduced by previous adaptive methods, its novelty lies in the introduction of a context-aware adaptive penalty mechanism. This is a fundamental departure from conservative, context-agnostic penalty. These contributions distinguish MOBA as a novel approach rather than a simple extension.
>
> Q2: The s baselines are exclusively from before 2023.
>
> Our original choice of baselines (MOPO, MAPLE, MOBILE) was deliberate to isolate the algorithmic contribution of our adaptive penalty by using identical underlying model-learning architectures. However, to demonstrate SOTA competitiveness, we have now benchmarked MOBA against MOREC (ICLR 2024) and ADMPO (ICLR 2025). As shown in our Public Rebuttal, MOBA remains competitive or superior to these recent methods, demonstrating that our contribution is orthogonal and additive to recent model-learning advancements.

---

### Official Review · Reviewer_dpkc · 2025-10-28

**Soundness:** 2
**Presentation:** 3
**Contribution:** 2
**Rating:** 4
**Confidence:** 3

**Summary:**

The paper introduces MOBA, a novel model-based offline RL approach. It addresses the limitation of traditional methods, which use fixed conservative penalties that overly restrict policy exploration. MOBA employs an adaptive contextual penalty that dynamically adjusts conservatism based on trajectory history, resulting in SOTA results and a tighter theoretical lower bound on the return.

**Strengths:**

1) Well motivated. It addresses the limitation of traditional methods, which use fixed conservative penalties that overly restrict policy exploration.
2) Both empirical and theoretical results are provided. Empirical results show strong improvement compared to baselines

**Weaknesses:**

1) The submission is targeting ICLR 2026, yet the baselines are exclusively from 2022–2023. There is a significant risk of lacking comparison to more recent, state-of-the-art methods.
2) The proposed method explicitly aims to solve issues in out-of-support regions, but the current experiments are performed solely on seen tasks. To properly validate the method's contribution, the authors must evaluate performance on unseen tasks to demonstrate robustness in more severely out-of-support regions.

**Questions:**

See weaknesses.

---

> ### Author Response · Authors · 2025-11-17
>
> We thank Reviewer dpkc for recognizing that our work is "well-motivated" and provides "strong improvement" with both empirical and theoretical results.
>
> W1: The submission is targeting ICLR 2026, yet the baselines are exclusively from 2022-2023.
>
> This is a fair and important concern. Our original choice of baselines (MOPO, MAPLE, MOBILE) was deliberate to isolate the algorithmic contribution of our adaptive penalty by using identical underlying architectures. However, to demonstrate SOTA competitiveness, we have now benchmarked MOBA against MOREC (ICLR 2024) and ADMPO (ICLR 2025). As shown in our Public Rebuttal. Results confirm that our adaptive penalty mechanism remains state-of-the-art.
>
> | Task            | MOPO | MOBILE | MAPLE | MOBA | ADMPO (ICLR2025) |
> |-----------------|------|--------|-------|------|--------------|
> | halfcheetah-rnd | 38.5 | 39.3   | 38.4  | 38.3 | 45.4 ± 2.8   |
> | hopper-rnd      | 31.7 | 31.9   | 10.6  | 33.2 | 32.7 ± 0.2   |
> | walker-rnd      | 7.4  | 17.9   | 21.7  | 24.1 | 22.2 ± 0.2   |
> | halfcheetah-med | 73.0 | 74.6   | 50.4  | 79.8 | 72.2 ± 0.6   |
> | hopper-med      | 62.8 | 106.6  | 21.1  | 105.6| 107.4 ± 0.6  |
> | walker-med      | 84.1 | 87.7   | 56.3  | 82.2 | 95.5 ± 8.7   |
> | halfcheetah-med-rep | 72.1 | 71.7 | 59.0  | 69.7 | 67.6 ± 3.4   |
> | hopper-med-rep  | 103.5| 103.9  | 87.5  | 110.8| 104.4 ± 0.4  |
> | walker-med-rep  | 85.6 | 89.9   | 76.7  | 95.1 | 95.6 ± 2.1   |
> | **Average**     | 62.1 | 69.3   | 46.9  | 72.5 | 71.4         |
>
> | Task Name     | MOPO | MOBILE | MOBA (Ours) | MOREC(ICLR2024) |
> |---------------|------|--------|-------------|------------|
> | HalfCheetah-L | 40.1 | 54.7   | 51.3 ± 0.4  | 53.5 ± 0.6 |
> | Hopper-L      | 6.2  | 17.4   | 33.0 ± 0.3  | 25.4 ± 1.3 |
> | Walker2d-L    | 11.6 | 37.6   | 70.5 ± 0.8  | 65.0 ± 1.3 |
> | HalfCheetah-M | 62.3 | 77.8   | 86.2 ± 1.3  | 84.1 ± 0.5 |
> | Hopper-M      | 1.0  | 51.1   | 74.6 ± 18.3 | 83.5 ± 3.8 |
> | Walker2d-M    | 39.9 | 62.2   | 78.6 ± 2.2  | 76.6 ± 1.7 |
> | **Average** | 26.9 | 50.1 | 65.7        | 64.7       |
>
> W2: Current experiments are performed solely on seen tasks.
>
> We would like to clarify our perspective on OOD and, in light of your feedback, we have also conducted new supplementary experiments to address this point directly.
>
> First, we respectfully argue that the standard offline RL benchmarks (like d4rl and NeoRL) are inherently OOD problems. The core challenge is to learn a policy from a static dataset that generalizes to "out-of-support" (OOS) state-action pairs during evaluation—that is, states not covered by the behavior policy. Our strong performance in Tables 1 & 2 already demonstrates MOBA's superior ability to handle this primary form of distribution shift.
>
> However, we understand your question is also aimed at a more explicit test of OOD, such as generalizing to new or unseen dynamics. To address this concern, we conducted a new set of experiments in a challenging discrete-action domain, "Hard-CartPole". Designation and details of the experiment can be find in our public official comment. Also, we list the details below for your convenience.
>
> Environment: We use the classical CartPole environment but significantly increase its difficulty by introducing velocity-proportional damping and real-world stochasticity.
>
> OOD Setup: To create a robust test of dynamic generalization, the collected trajectories are perturbed with varying pole lengths, pole masses, and cart masses, as detailed in the "Hard-CartPole Parameters" table below:
>
> | Description           | masscart (kg) | masspole (kg) | length (m) |
> |------------------|---------------|---------------|---------------|
> | Standard parameters | 1.0           | 0.1           | 0.5           |
> | Light pole        | 1.0           | 0.05          | 0.5           |
> | Heavy pole        | 1.0           | 0.15          | 0.5           |
> | Short light pole  | 1.0           | 0.08          | 0.4           |
> | Long heavy pole   | 1.0           | 0.12          | 0.6           |
>
> The behavior policy in this environment achieves an average score of 600.
>
> Results: In this new "Hard-CartPole" benchmark, MOBA achieves a score of 556, significantly outperforming all baselines, as shown in the performance table:
>
> | Task Name | MOPO | MAPLE | MOBILE | MOBA |
> | :--- | :--- | :--- | :--- | :--- |
> | Hard-CartPole | 421 | 421 | 493 | 556 |
>
> These supplementary results provide direct evidence that MOBA's adaptive contextual penalty allows it to better generalize across different dynamics, demonstrating superior OOD robustness

---

> > ### Comment · Reviewer_dpkc · 2025-11-26
> >
> > The authors' reply mentioned the aim to 'isolate the algorithmic contribution of our adaptive penalty by using identical underlying architectures.' Could the authors clarify the differences between MOREC and ADMPO and the other baselines in terms of settings and methodologies? In particular, why did the authors decide not to include these two baselines in the original submission? This is especially important given that the newly added baselines significantly outperform the old ones, which made me suspect the fair comparison on baselines proposed in year 2024-2026.

---

> > > ### Author Response · Authors · 2025-11-28
> > > **Reply to Reviewer dpkc: Clarification on Baselines**
> > >
> > > We thank Reviewer dpkc for the prompt follow-up and the fair question regarding our initial baseline selection. We appreciate the opportunity to clarify the methodological differences and the rationale behind our experimental design.
> > >
> > > We fully agree with your assessment that comparing against the most recent state-of-the-art methods is necessary to validate the competitiveness of our approach. We have incorporated ADMPO (ICLR 2025) and MOREC (ICLR 2024) into our experiments. As shown in the updated tables in our previous comment (and the revised paper), MOBA consistently outperforms or matches these strong baselines. These results confirm that our method remains state-of-the-art even against the latest advancements.
> > >
> > > Our decision to originally compare primarily against MOPO, MOBILE, and MAPLE was not intended to avoid stronger baselines, but rather to ensure a fair comparison to isolate the source of improvement.  MOBA utilizes the same standard probabilistic ensemble dynamics model as MOPO and MOBILE. By keeping the model learning component identical to these established baselines, we could demonstrate that the performance gains are strictly attributable to our proposed adaptive contextual penalty mechanism, rather than a more powerful underlying dynamics model.
> > >
> > > o answer your specific question on the settings and methodologies, ADMPO and MOREC introduce orthogonal improvements to the model learning and data selection phases, whereas MOBA focuses on the policy optimization phase.  ADMPO proposes the Any-step Dynamics Model (ADM). Instead of the standard 1-step prediction, it aims to directly predict future states $s_{t+h}$ from $s_t$. This effectively reduces bootstrapping and accumulated prediction error during model rollout. MOREC learns a Reward-Consistent Dynamics Reward Function. It acts as a transition filter, selecting synthetic transitions that are most consistent with the reward signal in the offline data to improve model quality.  In contrast, MOBA achieves its performance without changing the fundamental model learning objective (like ADMPO) or filtering data based on reward consistency (like MOREC). The fact that MOBA surpasses these methods while using a standard, simpler dynamics model highlights the efficacy and robustness of our adaptive penalty design.
> > >
> > > We hope this clarifies that our original submission aimed to showcase the specific contribution of the penalty mechanism. However, your suggestion to include these broader comparisons was entirely correct, and adding them has significantly strengthened the paper's claims.

---

> > > > ### Comment · Reviewer_dpkc · 2025-11-28
> > > >
> > > > I have no further questions. Although I am not fully convinced by the reply for not incorporating more modern methods, the newly provided experiments have addressed my main concerns and demonstrate the SOTA performance of the proposed method.
> > > >
> > > > By the way, in this stage, I can not change the rating to 6.

---

> > > > > ### Author Response · Authors · 2025-12-01
> > > > >
> > > > > We sincerely thank Reviewer dpkc for the prompt follow-up and for confirming that the supplementary experiments have addressed your main concerns.
> > > > >
> > > > > We are encouraged by your recognition of MOBA’s SOTA performance against the latest baselines. We completely understand the current system limitations regarding the rating update. We are nonetheless grateful that your constructive feedback prompted us to include comparisons with latest SOTA performance, as these results have significantly strengthened the empirical evidence of our paper.
> > > > >
> > > > > Tank you again for the time and effort you dedicated to reviewing our work.

---

### Official Review · Reviewer_hBaB · 2025-10-30

**Soundness:** 3
**Presentation:** 3
**Contribution:** 3
**Rating:** 6
**Confidence:** 3

**Summary:**

This paper focused on offline model-based RL. The authors proposed MOBA that introduces a context-aware penalty adaptation mechanism to dynamically adjust conservatism based on trajectory history. Theoretically, MOBA was proven to maximize a tighter lower bound on the true return. And it was shown to outperform state-of-the-art methods on benchmark tasks, highlighting the importance of adaptive uncertainty estimation in model-based offline RL. The authors provided theoretical justification and comprehensive experimental evidence.

**Strengths:**

1.The paper thoroughly identifies flaws in existing model-based offline RL methods (e.g., MOPO’s fixed penalties, MAPLE’s residual conservative bias) and proposes MOBA’s adaptive mechanism (integrating ε(s,a) and ω(s,a)) to resolve the "over-constraint vs. error accumulation" dilemma.

2.MOBA targets real scenarios (robotics, healthcare) by using dynamics ensembles to boost data efficiency, achieves top average scores on the near-real-world NeoRL benchmark, and provides detailed implementation details for reproducibility and deployment.

3.The paper follows a rigorous structure (from problem formulation to future directions) with tight links between theory and experiments, and uses precise terminology to explain complex concepts clearly, balancing rigor and readability.

**Weaknesses:**

1.The rationale behind the specific designs of ε(s, a) and ω(s, a) requires clarification. Please explain the motivation for formulating ε(s, a) as an entropy measure and ω(s, a) as the ensemble variance (or average covariance).

2.Experimental Section: Why is there no analysis on Out-of-Distribution (OOD) generalization and distribution shift?

3.Ablation Study: Is it need to add ablation results that analyze the runtime performance?

4.Variance details in the experimental section: What is the variance of the benchmark? How many times were the specific experiments conducted, and how were the parameters set (these may be available in the code)?

**Questions:**

Please refer to Weaknesses.

---

> ### Author Response · Authors · 2025-11-17
>
> We thank Reviewer hBaB for highlighting the strengths of our work, including our analysis of existing flaws, targeting of real-world scenarios, and rigorous structure. We address your specific concerns below.
>
> W1: The rationale behind the specific designs of $\epsilon(s,a)$ and $\omega(s,a)$ requires clarification.
>
> We elaborate on the intuition and rationale behind our selection of the two proxies in the public official comment. Below, we give a short and intuitive explanation.
>
> $\epsilon$ (Entropy): We use (1 - normalized entropy) as a proxy for context confidence.  $\epsilon$ captures the ambiguity of the probing phase.  A high-quality, low-error context $z_t$ from the extractor leads to a confident, low-entropy policy. A poor, high-error context $z_t$ leads to an uncertain, high-entropy policy
>
> $\omega$ (Variance Ratio): We use the ratio of inter-model variance (disagreement) to intra-model variance (known uncertainty). A high ratio indicates high disagreement relative to known uncertainty, which is a strong signal of a large dynamics-gap (i.e., the ensemble does not cover the true dynamics)
>
> W2: Why is there no analysis on Out-of-Distribution (OOD) generalization and distribution shift
>
> Thank you for this crucial question. We would like to clarify our perspective on OOD and, in light of your feedback, we have also conducted new supplementary experiments to address this point directly.
>
> First, we respectfully argue that the standard offline RL benchmarks (like d4rl and NeoRL) are inherently OOD problems. The core challenge is to learn a policy from a static dataset that generalizes to "out-of-support" (OOS) state-action pairs during evaluation—that is, states not covered by the behavior policy. Our strong performance in Tables 1 & 2 already demonstrates MOBA's superior ability to handle this primary form of distribution shift.
>
> However, we understand your question is also aimed at a more explicit test of OOD, such as generalizing to new or unseen dynamics. To address this concern, we conducted a new set of experiments in a challenging discrete-action domain, "Hard-CartPole".
>
> Environment: We use the classical CartPole environment but significantly increase its difficulty by introducing velocity-proportional damping and real-world stochasticity.
>
> OOD Setup: To create a robust test of dynamic generalization, the collected trajectories are perturbed with varying pole lengths, pole masses, and cart masses, as detailed in the "Hard-CartPole Parameters" table below:
>
> | Description           | masscart (kg) | masspole (kg) | length (m) |
> |------------------|---------------|---------------|---------------|
> | Standard parameters | 1.0           | 0.1           | 0.5           |
> | Light pole        | 1.0           | 0.05          | 0.5           |
> | Heavy pole        | 1.0           | 0.15          | 0.5           |
> | Short light pole  | 1.0           | 0.08          | 0.4           |
> | Long heavy pole   | 1.0           | 0.12          | 0.6           |
>
> The behavior policy in this environment achieves an average score of 600.
>
> Results: In this new "Hard-CartPole" benchmark, MOBA achieves a score of 556, significantly outperforming all baselines, as shown in the performance table:
>
> | Task Name | MOPO | MAPLE | MOBILE | MOBA |
> | :--- | :--- | :--- | :--- | :--- |
> | Hard-CartPole | 421 | 421 | 493 | 556 |
>
> These supplementary results provide direct evidence that MOBA's adaptive contextual penalty allows it to better generalize across different dynamics, demonstrating superior OOD robustness even in discrete-action settings. We will add these new findings and the environment details to the Appendix of our revised paper.
>
> W3: Is it need to add ablation results that analyze the runtime performance?
>
> The primary computational overhead of MOBA comes from the recurrent context extractor, which is shared with baselines like MAPLE. The additional cost of calculating $\epsilon$ (policy entropy) and $\omega$ (ensemble variance) involves only lightweight algebraic operations and is minimal. To further illustrate this, we present a comparative analysis of the running-time complexity between MOBA and commonly employed model-based algorithms. we measure the runtime per epoch (1K gradient steps) and the number of parameters for each algorithm based on the hopper-medium-v2 task.
>
> | Method | Runtime (s/epoch) | Number of parameters |
> |--------|-------------------|----------------------|
> | MOPO   | 7                 | 2.2M                 |
> | MOBILE | 8                 | 2.2M                 |
> | MOBA   | 7                 | 2.2M                 |
>
> W4: Variance details in the experimental section. You are correct, this was an oversight. All results reported in Table 1 and Table 2  are the mean and standard deviation over 5 random seeds. We will add this crucial detail to the main paper in Section 5.1 and to the hyperparameter tables in Appendix B.

---

### Official Review · Reviewer_eeS2 · 2025-10-31

**Soundness:** 3
**Presentation:** 2
**Contribution:** 2
**Rating:** 4
**Confidence:** 3

**Summary:**

This paper proposes a model-based offline RL method that incorporates context awareness into the mainstream conservative model-based approaches (e.g., MOPO-style). Compared to the fixed model uncertainty penalty used in MOPO, the proposed framework scales the model uncertainty penalty adaptively by context, so the policy is penalized more in out-of-distribution regions and less where both the policy and model are confident. Under their assumptions, this yields a tighter lower bound on return than a fixed penalty, as the adaptive penalty is used to avoid excessive pessimism. Experiments on standard benchmarks support these claim.

**Strengths:**

1. The motivation is natural: context-agnostic pessimism (i.e., a fixed penalty) can be overly conservative, which constrains exploration in out-of-support regions. The paper addresses this by introducing a context-aware penalty term in the reward function.

2. The flow of presentation is organized: the method section begins with a brief introduction to a probing-reduction paradigm (Section 4.1), which motivates the proposed scalar modulator on the traditional uncertainty-penalization term. Discussions in section 4.1 also motivates the proof of Lemma 4.1.

3. Empirical support: Experiments on standard benchmarks demonstrate the effectiveness of the proposed context aware uncertainty penalty.

**Weaknesses:**

My main concerns are as follows:

1. Ambiguity in theoretical notation and definitions: in section 4.2, the definition of $\hat{T}$ is not that clear. It appears to denote the ensemble transition in the proof, whereas section 4.1 defines it as the single best model. This mismatch makes Lemma 4.1 hard to parse and invites confusion when comparing to MOPO’s Lemma 4.1. In addition, definitions of $\lambda$, $\epsilon$, $\omega$ in Lemma 4.1 are lacking. I suggest to revise the presentation in Lemma 4.1.

2. Vague rationales of $\epsilon$, $\omega$ used in section 4.3: In section 4.3, the paper selects specific proxies for them but it is unclear why these choices follow from the theory. I suggest to connect these proxies to the constructs derived in Appendix A.4.

**Questions:**

1. Could you explicitly derive how the proxies in section 4.3 correspond to the theoretical constructs in Appendix A.4?

2. How do you ensure $\epsilon$ and $\omega$ are both less than 1 in practice? Could you discuss a bit more why $\epsilon$ and $\omega$ in section 4.3 satisfy this condition?

---

> ### Author Response · Authors · 2025-11-17
> **Response to Reviewer eeS2**
>
> We thank Reviewer eeS2 for the positive assessment of our motivation, presentation flow, and empirical support. We address your specific concerns below.
>
> W1: Ambiguity in theoretical notation ($\hat{T}$ definition) and missing definitions in Lemma 4.1.
>
> We apologize for the inconsistent use of $\hat{T}$. In the revised version, we will strictly define $T = \{T_i\} $  as the full ensemble, $T_\phi$ as the ensemble average (as used in Appendix A.1 ), and $\hat{T}_\theta$ as the single "closest" model. We will correct this throughout Section 4.1, 4.2, and Appendix A to ensure clarity.
> The definitions for $\lambda, \epsilon, \omega$ were indeed placed after Lemma 4.1. In our revision, we will add a concise, high-level definition of these terms before stating the lemma to improve readability, with the full derivation following.
>
> W2, Q1: Vague rationale for $\epsilon, \omega$ proxies and their connection to Appendix A.
>
> We elaborate on the intuition and rationale behind our selection of the two proxies in the public official comment. Below, we provide an intuitive explanation of their explicit connection to the theoretical constructs outlined in Appendix A.4.
>
> For $\epsilon(s,a)$ (Context-Recognition Error): Our proxy is $\epsilon = 1 - H(\pi_\theta(\cdot|s_t, z_t))$. The theoretical construct in Appendix A, $\epsilon(s,a)$, is the "estimated probability that $j \ge N_p$" (i.e., that we are past the probing phase). Our logic is as follows:If the context extractor $\psi_\xi$ fails (i.e., high recognition error, $j < N_p$), the context $z_t$ is uninformative or noisy.A rational policy $\pi(a|s,z)$, when given an uninformative $z_t$, cannot commit to a single optimal action and will thus exhibit high entropy (i.e., $\epsilon$ is low).Conversely, if the extractor succeeds (low error, $j \ge N_p$), $z_t$ is highly informative, allowing the policy to be confident and exhibit low entropy (i.e., $\epsilon$ is high).Thus, (1 - Policy Entropy) is a direct and principled proxy for the "context confidence" or the probability of being in the "reduced" phase.
>
> For $\omega(s,a)$ (Dynamics-Gap Error): Our proxy is $\omega(s_t, a_t) = \frac{Var_i[\mu_i(s_t, a_t)]}{\mathbb{E}[|\Sigma_i(s_t, a_t)|]}$. We formulate $\omega$ as the ratio $\frac{\text{Provided Diversity}}{\text{Required Diversity}}$. This proxy measures whether the ensemble is rich enough to cover the true dynamics. The Numerator (Ensemble Variance) represents the diversity of dynamics provided by our current models. The Denominator (Aleatoric Uncertainty) represents the diversity required to account for the inherent stochasticity of the real environment. When this ratio is large (i.e., Provided > Required), it indicates that the ensemble's divergence is sufficient to cover the true world dynamics within its support. Conversely, a small ratio suggests that the ensemble is too narrow to capture the real world's inherent complexity, signaling a risk of over-confidence.
>
> Q2: How do you ensure $\epsilon$ and $\omega$ are both less than 1?
>
> For $\epsilon$: We use normalized entropy, ensuring $H \in [0,1]$, so $\epsilon \in [0,1]$.
>
> For $\omega$:  In our implementation, we clip $\omega$ to $[0, 1]$. Theoretically, when the "Provided Diversity" (numerator) fully exceeds the "Required Diversity" (denominator), the ratio approaches or exceeds 1. At this point, we conclude that the ensemble fully covers the real world, so we clip the value to maintain the bound required by Theorem 4.1.
>
> We hope these clarifications address your concerns and will work to integrate them clearly into the revised manuscript.

---

### Author Response · Authors · 2025-11-19

We thank all reviewers for their constructive feedback. We are encouraged by the recognition of our well-motivated problem setting , rigorous structure , and empirical support. Below, we address the three common questions raised across reviews.

Comparison with New SOTA Baselines (Response to dpkc, sBH5)

Reviewers noted that our baselines were primarily from 2022-23. We initially selected these works because our experimental setup for environment model learning is identical to theirs, thereby enabling a more direct and focused demonstration of the algorithmic advantages introduced in our approach.  However, to demonstrate MOBA's current relevance, we have added comparisons against MOREC[1] (ICLR 2024)  and ADMPO[2] (ICLR 2025). Results show that MOBA remains highly competitive, outperforming latest SOTA on neorl and d4rl tasks

| Task            | MOPO | MOBILE | MAPLE | MOBA(Ours) | ADMPO (ICLR2025) |
|-----------------|------|--------|-------|------|--------------|
| halfcheetah-rnd | 38.5 | 39.3   | 38.4  | 38.3 | 45.4 ± 2.8   |
| hopper-rnd      | 31.7 | 31.9   | 10.6  | 33.2 | 32.7 ± 0.2   |
| walker-rnd      | 7.4  | 17.9   | 21.7  | 24.1 | 22.2 ± 0.2   |
| halfcheetah-med | 73.0 | 74.6   | 50.4  | 79.8 | 72.2 ± 0.6   |
| hopper-med      | 62.8 | 106.6  | 21.1  | 105.6| 107.4 ± 0.6  |
| walker-med      | 84.1 | 87.7   | 56.3  | 82.2 | 95.5 ± 8.7   |
| halfcheetah-med-rep | 72.1 | 71.7 | 59.0  | 69.7 | 67.6 ± 3.4   |
| hopper-med-rep  | 103.5| 103.9  | 87.5  | 110.8| 104.4 ± 0.4  |
| walker-med-rep  | 85.6 | 89.9   | 76.7  | 95.1 | 95.6 ± 2.1   |
| **Average**     | 62.1 | 69.3   | 46.9  | 72.5 | 71.4         |

| Task Name     | MOPO | MOBILE | MOBA (Ours) | MOREC(ICLR2024) |
|---------------|------|--------|-------------|------------|
| HalfCheetah-L | 40.1 | 54.7   | 51.3 ± 0.4  | 53.5 ± 0.6 |
| Hopper-L      | 6.2  | 17.4   | 33.0 ± 0.3  | 25.4 ± 1.3 |
| Walker2d-L    | 11.6 | 37.6   | 70.5 ± 0.8  | 65.0 ± 1.3 |
| HalfCheetah-M | 62.3 | 77.8   | 86.2 ± 1.3  | 84.1 ± 0.5 |
| Hopper-M      | 1.0  | 51.1   | 74.6 ± 18.3 | 83.5 ± 3.8 |
| Walker2d-M    | 39.9 | 62.2   | 78.6 ± 2.2  | 76.6 ± 1.7 |
| **Average** | 26.9 | 50.1 | 65.7        | 64.7       |

New OOD Experiment: "Hard-CartPole" (Response to hBaB, dpkc)

To address concerns about Out-of-Distribution (OOD) robustness beyond standard benchmarks, we introduced a "Hard-CartPole" environment with unseen physics (varying pole length from 0.4 to 0.6, masses from 0.05 to 0.15).

 The offline dataset was collected exclusively using "Standard" physical parameters . This mimics the scenario where the agent learns from historical data covering only nominal dynamics. The learned policy was deployed and evaluated in environments with perturbed, unseen parameters.

Results: In this new "Hard-CartPole" benchmark, MOBA achieves a score of 556, significantly outperforming all baselines, as shown in the performance table:

| Task Name | MOPO | MAPLE | MOBILE | MOBA |
| :--- | :--- | :--- | :--- | :--- |
| Hard-CartPole | 421 | 421 | 493 | 556 |

Rationale for Proxies $\epsilon$ and $\omega$ (Response to eeS2, hBaB, sBH5)

$\epsilon$ captures the ambiguity in the probing phase. When the context extractor fails to eliminate the incompatible models, the agent effectively faces a superposition of multiple possible dynamics. Since the optimal action often varies significantly across these conflicting dynamics, a rational policy cannot commit to a single action. Instead, it must maintain a dispersed probability mass (high entropy) to hedge against this uncertainty. Therefore, high entropy is not merely randomness; it is a direct signature of the agent facing multiple possible environments due to an uninformative context.

We formulate $\omega$ as the ratio $\frac{\text{Provided Diversity}}{\text{Required Diversity}}$. This proxy measures whether the ensemble is rich enough to cover the true dynamics. The Numerator (Ensemble Variance) represents the diversity of dynamics provided by our current models. The Denominator (Aleatoric Uncertainty) represents the diversity required to account for the inherent stochasticity of the real environment. When this ratio is large (i.e., Provided > Required), it indicates that the ensemble's divergence is sufficient to cover the true world dynamics within its support. Conversely, a small ratio suggests that the ensemble is too narrow to capture the real world's inherent complexity, signaling a risk of over-confidence.




[1] Luo, Fan-Ming, et al. "Reward-Consistent Dynamics Models are Strongly Generalizable for Offline Reinforcement Learning." The Twelfth International Conference on Learning Representations.

[2] Lin, Haoxin, et al. "Any-step Dynamics Model Improves Future Predictions for Online and Offline Reinforcement Learning." The Thirteenth International Conference on Learning Representations.

---

### Meta-Review · Area_Chair_XxZJ · 2026-01-07

**Summary:**

The paper considers the problem of model-based offline reinforcement learning (MBRL) and argue that "static" MBRL approach prevents policies from exploiting regions where the model might be locally accurate despite general uncertainty. A method named MOBA (Model-Based Offline RL with Adaptive Contextual Penalties) is proposed which introduces a dynamic penalty coefficient. This coefficient scales the standard uncertainty penalty based on two factors: the confidence of the policy in the current context (estimated via action entropy) and the discrepancy between the ensemble models (estimated via a ratio of variance to expected covariance).

The most critical issue with the paper (as mentioned by Reviewer sBH5 and Reviewer eeS2) is the lack of technical soundness and weak justification for the practical proxies used to satisfy the theoretical bounds. Specifically, the choice of two key parameters $\epsilon$ and $w$ is weak and unclear. The rebuttal was unable to clarify this. Additionally, the paper initially missed multiple important baselines of last 3-4 years. Some of these were added in the rebuttal period but the performance of proposed approach is not significantly better. I believe they should have been in the paper from the beginning. Overall, there are many changes that needs to be made for this paper to be accepted at the conference. Therefore, I recommend rejecting while requesting the authors to incorporate comments from reviewers in the future submission.

**Reviewer Concerns:**

Major concerns about the two parameters and baselines were not addressed adequately.

**Reviewer Scores:**

Reviewer dpkc agreed to increase the score based on the new experiments. I believe reviewer sBH5 would have maintained their score.

---

### Decision · Program_Chairs · 2026-01-26

Reject